# Development of Real-Time Implementation of a Wind Power Generation System with Modular Multilevel Converters for Hardware in the Loop Simulation Using MATLAB/Simulink

**Dong-Cheol Shin and Dong-Myung Lee \***

School of Electronic and Electrical Engineering, Hongik University, Seoul 04066, Korea
* Correspondence: dmlee@hongik.ac.kr

**Abstract:** In this study, we propose a wind power generation system model for operating modular multilevel converter (MMC) in a hardware-in-the-loop simulation (HILS) application. The application of the MMC is a system that connects wind power to a grid through high-voltage direct current (HVDC) in the form of back-to-back connected MMCs, whereas a HILS is a system used to test or develop hardware or a software algorithm with real time. A real-time operation model of the MMC is required to conduct a HILS experiment. Although some studies have introduced the HILS model of MMCs for grid connection using PSCAD/EMTDC, it is difficult to find a study in the literature on the model using Matlab/Simulink, which is widely used for power electronic simulation. Hence, in this paper, we propose a real-time implementation model employing a detailed equivalent model (DEM) using MATLAB/Simulink. The equivalent model of both wind power generation system and MMC are presented in this paper. In addition, we describe how to implement components such as a variable resistor that is not provided in the Simulink's library. The feasibility of the proposed model is demonstrated with real-time operation of a wind power generation system.

**Keywords:** hardware-in-the-loop simulation (HILS); modular multilevel converter (MMC); Simulink model; detailed equivalent circuit; wind power generation

## 1. Introduction

Over the past two decades there has been growing interest in renewable sources of energy, which have the advantages of eliminating harmful gas emissions and providing infinite resources of primary energy. Wind power generation has the advantage of using vast resources of wind power and it attracts attention as a clean energy that can minimize the emission of harmful substances [1–4]. Wind power generation depends on weather conditions, is concentrated, and gets larger in suitable geographical locations. To transmit a large amount of power, an efficient transmission system is required. To this end, there are two types of grid connection systems for wind power generation, called high-voltage alternating current (HVAC) and high-voltage direct current (HVDC).

In the case of HVAC, as the transmission line becomes longer, the loss due to reactive power increases, and additional equipment is needed to compensate. On the other hand, HVDC requires only a pair of lines and has less loss of power. In addition, in the case of HVAC, a connection between power grids with different frequencies is not possible, while an HVDC link makes this possible [5–7]. Thanks to these merits, HVDC technology has been spotlighted in the field of high-voltage transmission, and research has been actively conducted on such aspects as system analysis, power conversion technology, and topology [8–10]. For power conversion in HVDC, modular multilevel converter (MMC) topology was proposed by R. Marquardt in 2003 [11]. A MMC maximizes the modularization

of the switching element of the converter. Because of the structure with the submodule (SM), the output voltage can be easily adjusted, and has the advantage of easy maintenance.

Since an MMC has a modular series-connected structure, the voltage stress of individual switching devices is low, and the alternating current (AC) side output waveform has a low total harmonic distortion (THD) even at a low switching frequency, thereby providing high quality output voltage [12–16]. For this reason, the voltage type HVDC in a commercial operation mainly uses an MMC structure [17–19]. When grids are connected with HVDC, output voltages at several hundred levels are generated. To this end, several hundred SMs are connected in series.

It is difficult to make and implement actual products to verify algorithms in terms of time and economy. Therefore, real-time simulations should be applied using hardware-in-the-loop simulation (HILS). HILS is a real-time simulation system that combines hardware and software and is a technique that can analyze complex models in real time. In recent years, HILS systems have frequently been used when the cost, time, and safety requirements for building a test environment matter, such as in power electronics [20,21].

Many techniques have been proposed to simulate MMCs in which several tens to several hundred SMs of the same shape are connected in series in real time [22–28]. The model introduced in [22,23] applies a traditional detailed insulated gate bipolar transistor (IGBT)-based modeling method. The detailed modeling of the IGBTs, which were used in [22,23] make it possible to reproduce the nonlinear behavior of multiple simultaneous multiple switching events and to account for power loss. In addition, there are advantages to apply various topologies such as detailed circuits and internal defects of SMs. This model has the advantage of enabling the application of various topologies such as faults due to internal flaws, because it can show the detailed circuit of the SM. However, this model requires a lot of computation, and therefore an expensive system is required.

The average model introduced in [24–26] ignores the details of the IGBT switches in an individual SM and only shows the ideal switching behavior. The model shows fast computational speed and a small number of computations but has the demerit that it does not represent the internal dynamics of the MMC. In the detailed equivalent circuit models (DEM) introduced in [27,28], the number of nodes in the circuit is reduced by replacing the switching element such as IGBT with on/off resistors. The DEM is well suited for a real-time simulation model for HILS because of sufficient accuracy and adequate simulation speed.

It should be mentioned that for the model in [27,28], PSCAD/EMTDC®software is used. However, in this paper, we propose a method of modeling MMCs applied with DEM using MATLAB/Simulink®(hereinafter Simulink) which is a tool widely used in development of power converter algorithms. Simulations of power converters using Simulink in PCs are extensively carried out. However, as far as we know, a wind power generation system model with a MMC applied, which are applied to HILS, has not been proposed in the literature. Therefore, in this study, we intend to show the model using Simulink which is widely used as an engineering tool. To verify the validity of the model, we show the results of HILS experiments in which a model of grid connection for wind power generation using MMCs was implemented in real time.

This paper is organized as follows: In Section 2, we describe the structure of a MMC; in Section 3, we explain the modeling of the HVDC system with the MMC for grid connection for wind power generation and show how to model the MMC system and wind power generation using Simulink; in Section 4, we describe the implementation of the HILS using OP4510, which is a real-time OS and explain the modeled wind power system, 31-level MMC for grid connection, and the controller using the FPGA, as well as describe the controller design using the FPGA and the state machine in controller implementation; and in Section 5, we summarize the experimental results of the HILS system with the proposed model to verify the validity of the proposed model.

## 2. Configuration of an MMC

Figure 1 illustrates the basic structure of a three-phase N+1 level MMC. It is similar to a three-phase two-level converter in that each phase has three legs with midpoint connections, which form the AC stage of the converter.

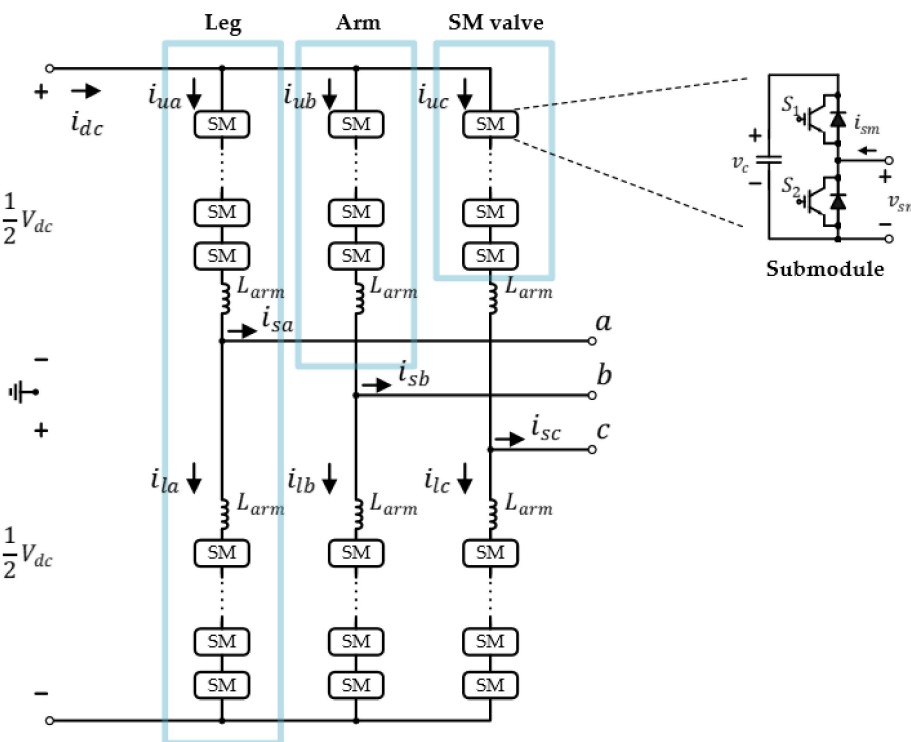

**Figure 1.** Schematic of a three-phase alternating current (AC) system connected with a modular multilevel converter (MMC).

One leg consists of two arms, which consists of SM valves and an arm inductor ($L_{arm}$). One arm has N pieces of SMs connected in series. In the case of the MMC in this research, each SM consists of a DC capacitor and two IGBTs, called $S_1$ and $S_2$. The two switches work in a complementary fashion under normal operating conditions. The output SM voltage ($v_{sm}$) becomes $v_c$, the capacitor voltage, when $S_1$ is ON, and $S_2$ is OFF. When bypassed ($S_1$ is OFF and $S_2$ is ON), $v_{sm}$ becomes 0. Therefore, the arm voltage, which is the summation of the output voltage of each SM, can be considered as a controllable voltage source.

## 3. Proposed Simulink Real-Time Model for the HILS Application

A diagram of the wind power generation system with grid connection using an MMC system, studied herein, is shown in Figure 2. The generator for the wind farm is a doubly fed induction generator (DFIG). The wind power generation system is connected to grids by the MMC. The proposed model is a real-time model for HILS application. The method to carry out the simulation is classified as a fixed time step method or a variable time step method. In this research, real-time simulations are carried out in the fixed time step. In the case of fixed time step simulations, the calculation should be completed in one step and the calculation result value should be obtained in the next step.

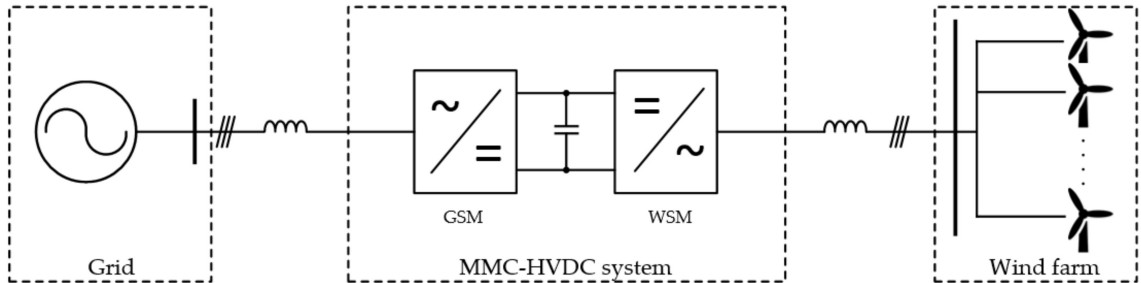

**Figure 2.** Diagram of the wind power generation system with grid connection by high-voltage direct current (HVDC) employing an MMC.

In cases where many switching elements are used, as with MMCs, a long calculation time is necessary due to the increase in the number of nodes. The number of switches recommended in OP4510, the real-time OS used in this study for real-time simulations, is limited to 60. The back-to-back connected MMC, shown in Figure 2, has 31 levels. Although a half-bridge type is used in the SM, 720 IGBTs are used for back-to-back MMC topology. The real-time simulation of a power converter with 720 switching elements is not possible. Therefore, an equivalent model is necessary for real-time implementation. Thus, a DEM that can dramatically reduce the number of nodes with the equivalent resistorization of switching elements was applied to modeling of MMCs. This is introduced in Section 3.1 and an equivalent model of wind power generation systems is explained in Section 3.2.

### 3.1. Detaild Equivalent Model (DEM) for the MMC

In this study, 30 serially connected SM blocks per arm were implemented with one equivalent circuit, which is based on the DEM proposed in [29]. The DEM in [29] was implemented with PSCAD/EMTDC. In addition, [29] did not introduce the implemented model. In this study, an implementation model using Simulink is shown in detail.

The SM block illustrated in Figure 3a is modeled as Figure 3b. In the DEM, the two switching elements $S_1$ and $S_2$ serve as a bidirectional switch in which the IGBT and the diode are connected in parallel. Since two switches operate complementarily, only one device is turned on and the device can be expressed as equivalent resistances of $r_1$ and $r_2$. The resistor has one of two values, $R_{on}$ or $R_{off}$, depending on the switching operation, where the $R_{on}$ value is several mΩ and the $R_{off}$ value is several MΩ.

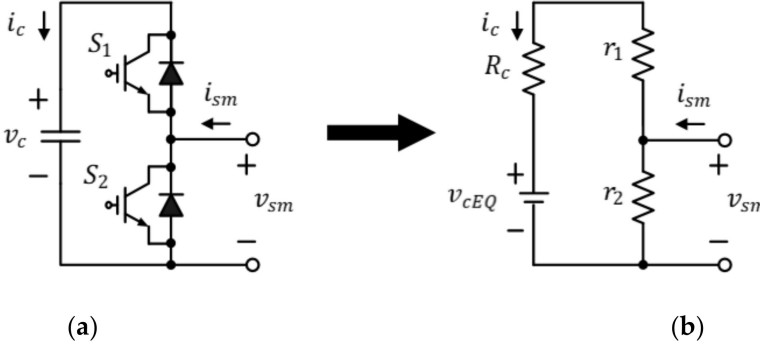

(**a**)                                      (**b**)

**Figure 3.** Submodule (SM) model for the detailed equivalent circuit. (**a**) SM circuit; (**b**) A resultant equivalent circuit for the SM.

The capacitor, as shown in Figure 3a, is modeled with an equivalent voltage source $v_{cEQ}$ and resistor ($R_c$) using the trapezoidal integration method. The SM capacitor voltage ($v_c$) is expressed as Equation (1), where $C$, $\Delta T$, and $i_c$ represent capacitance, calculation period, and the capacitor current, respectively. Meanwhile, $R_c$ is determined by the calculation period ($\Delta T$) and capacitance, as shown

in Equation (2), and $v_{cEQ}(t - \Delta T)$ is calculated with $R_c$, the $i_c(t - \Delta T)$ one step before the current time point of calculation, and the capacitor voltage $v_c(t - \Delta T)$ as Equation (3).

$$v_c(t) = R_c i_c(t) + v_{cEQ}(t - \Delta T) \tag{1}$$

$$R_c = \frac{\Delta T}{2C} \tag{2}$$

$$v_{cEQ}(t - \Delta T) = \frac{\Delta T}{2C} i_c(t - \Delta T) + v_c(t - \Delta T) \tag{3}$$

Figure 3b is implemented through Figure 4. The value to be obtained is $v_{sm}$. In Figure 4a, since $R_c$ and $r_1$ are serially connected to $v_{cEQ}$, source transformation is applied so that the circuit is transformed, as shown in Figure 4b, and since $r_2$ is connected in parallel to $r_1 + R_c$, the circuit is expressed as a Norton equivalent circuit, as shown in Figure 4c. If source transformation is again applied to the Norton equivalent circuit, a Thevenin equivalent circuit can be obtained, as shown in Figure 4d, and the output voltage of the SM, $v_{sm}$, is as shown in Equation (4). The equations for the resistance and voltage of the Thevenin equivalent circuit in Equation (4) can be obtained in the process of applying source transformation to the Norton equivalent circuit, and they are Equations (5) and (6), respectively.

$$v_{sm}(t) = r_{smEQ} i_{sm}(t) + v_{smEQ}(t - \Delta T) \tag{4}$$

$$r_{smEQ} = r_2 \left( 1 - \frac{r_2}{r_1 + r_2 + R_c} \right) \tag{5}$$

$$v_{smEQ}(t - \Delta T) = v_{cEQ}(t - \Delta T) \left( \frac{r_2}{r_1 + r_2 + R_c} \right) \tag{6}$$

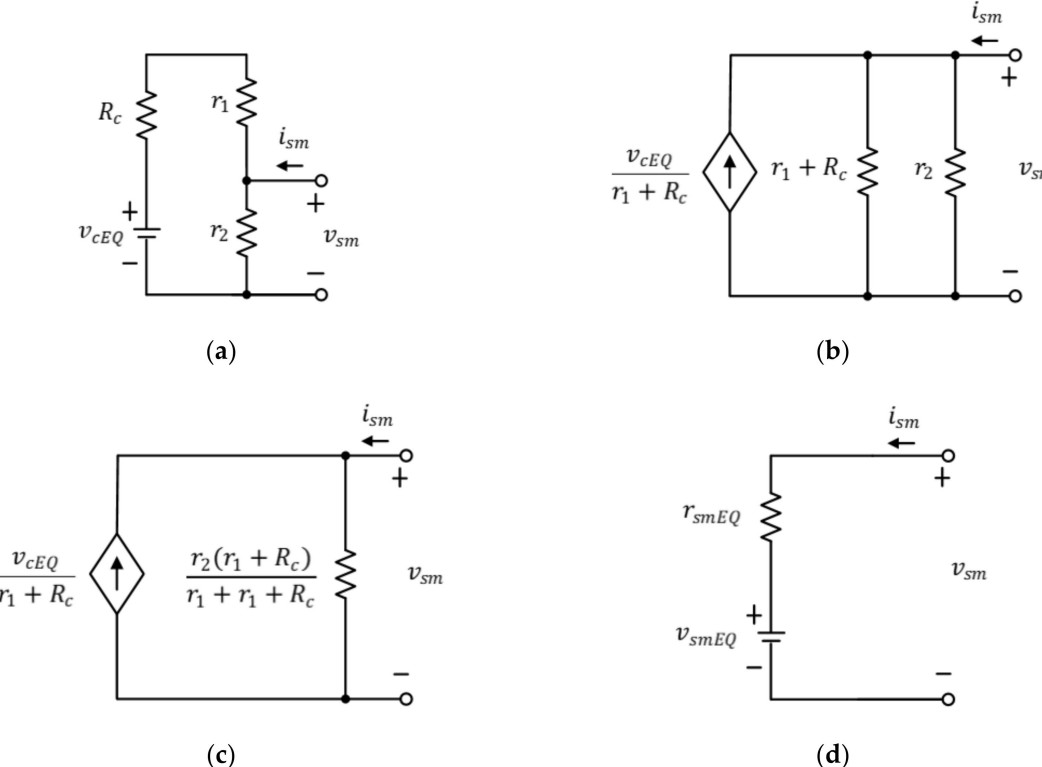

**Figure 4.** Thevenin equivalent circuit of SM. (**a**) An equivalent circuit for an SM; (**b**) After using source transformation; (**c**) Norton equivalent circuit for an SM; and (**d**) Thevenin equivalent circuit for an SM.

Figure 5 displays the Norton equivalent circuit to implement the equivalent circuit obtained as depicted in Figure 4 by using Simulink. Since the current of the SM in the same arm is equal to the arm current ($i_{arm}$), $i_{sm}$ can be referred to as $i_{arm}$. The output voltage of the SM valve ($v_{sm\_valve}$) is the sum of the output voltages of the individual SM ($v_{sm\_i}$), as Equation (7). Equations (8) and (9) can be obtained. Thus, the entire SM of an arm, which is connected in series, can be modeled as a Thevenin equivalent circuit or a Norton equivalent circuit, as shown in Figure 5. Here, the current ($i_{eq}$) of the dependent current source of the Norton equivalent circuit shown in Figure 5b can be obtained by applying source transformation to Figure 5a, as Equation (10).

$$v_{sm\_valve}(t) = \sum_{i=1}^{N} v_{sm(i)} \tag{7}$$

$$= \sum_{i=1}^{N} r_{smEQ(i)} i_{sm}(t) + \sum_{i=1}^{N} v_{smEQ(i)}(t - \Delta T) \tag{8}$$

$$= r_{eq} i_{arm}(t) + v_{eq} \tag{9}$$

$$i_{eq} = \frac{v_{eq}}{r_{eq}} \tag{10}$$

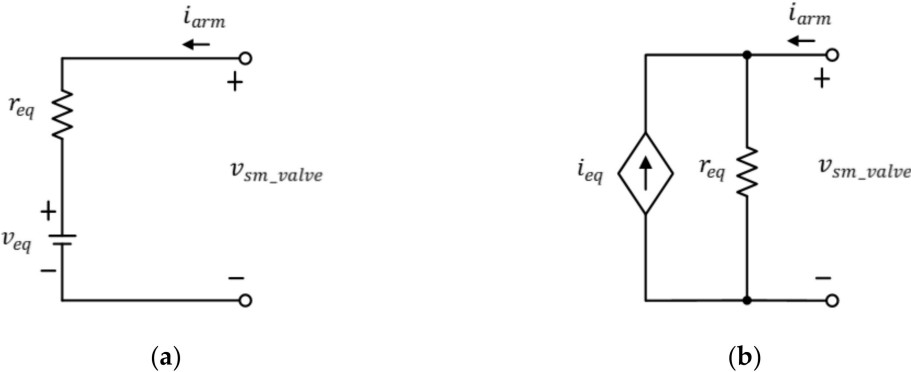

(**a**)          (**b**)

**Figure 5.** Equivalent circuit of the SM valve. (**a**) Thevenin equivalent circuit of the SM valve; (**b**) Norton equivalent circuit of the SM valve.

### 3.2. The Proposed DEM Implemented Using MATLAB/Simulink

Figure 6 depicts an overall simulation model of a wind power generation system, with grid connection using an MMC. The Simulink model of the phase leg of the MMC is shown in Figure 7a. Figure 7b,c shows the inside of an equivalent model of arm consisted of the SM valve. As shown in Figure 7b,c, the equivalent model of one arm has two blocks. In the first block, shown in Figure 7b, Equations (1) to (10) are implemented, that is, the voltages ($v_c$) of individual SM capacitors according to switching states, the resistance ($r_{eq}$) used in the Norton equivalent circuit of the SM valve, and the setpoint ($i_{eq}$) of the dependent current source are calculated. The second block, shown in Figure 7c, receives the $r_{eq}$ and $i_{eq}$ values calculated in the first block as inputs to actually generate the output voltage ($v_{sm\_valve}$) of the SM valve, that is, the second block implements the circuit shown in Figure 5b.

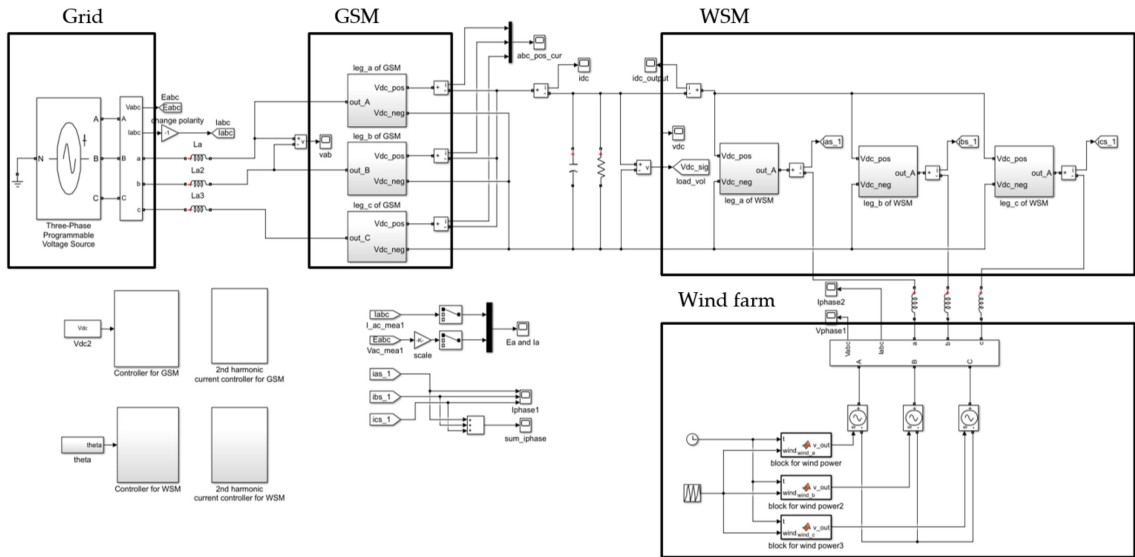

**Figure 6.** Overall simulation model of a wind power generation system with grid connection using back-to-back connected MMC developed using Simulink for the hardware-in-the-loop simulation (HILS) application.

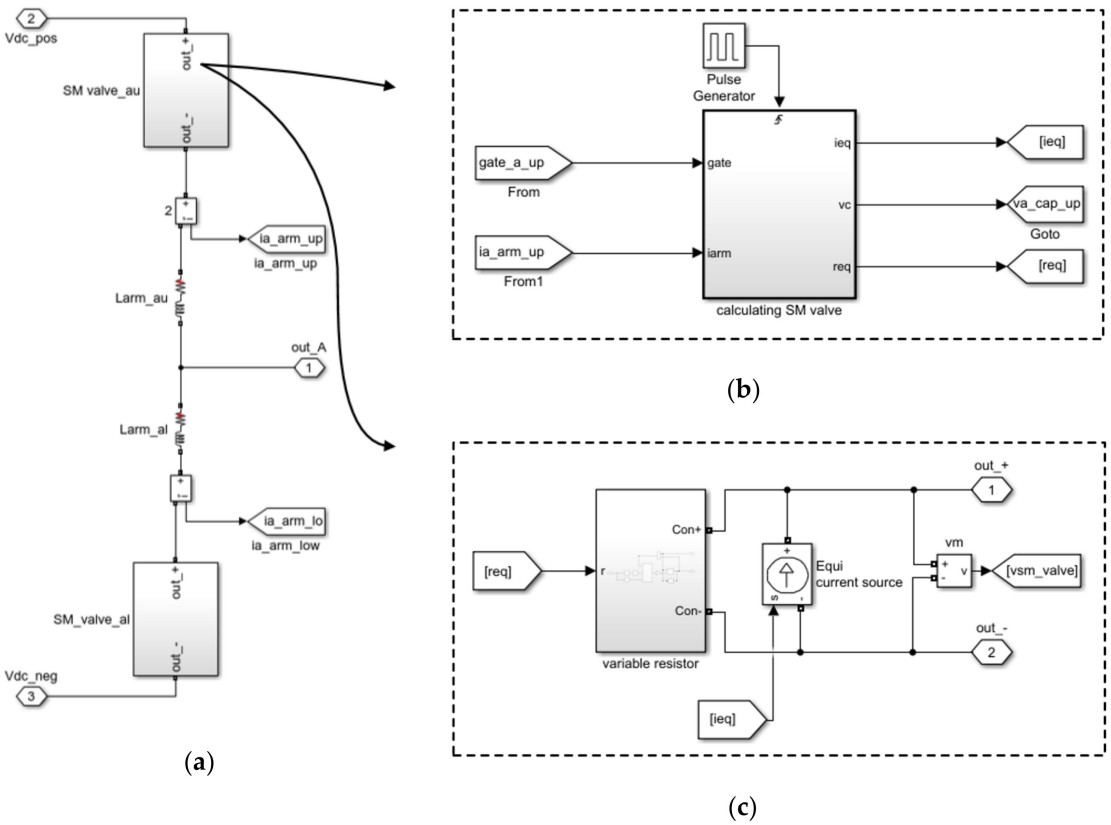

**Figure 7.** Simulink model of one phase leg of MMC. (**a**) Simulation model of one leg of MMC; (**b**) A block where the equivalent resistance and individual $v_c$ were implemented; (**c**) A block where the $v_{sm\_valve}$ was implemented using the Norton equivalent circuit.

Figure 8 is the internal block of Figure 7b,c. Figure 8a displays the inside of Figures 7b and 8b shows the block where the variable resistance of Figure 7c was implemented. Figure 8a consists of a trigger block and two Matlab function blocks (①, ②). These blocks carry out a calculation once per

10 µs with trigger signals. Matlab function block ① is in charge of the implementation of Equation (1), which is the voltages of individual SM capacitors. Since Equations (2) and (3) are necessary to implement Equation (1), gate signals (gate port), the arm current (iarm port), and the initial value of the SM capacitor (vc_init port) are received as default inputs to carry out calculations at a constant calculation cycle through the trigger block.

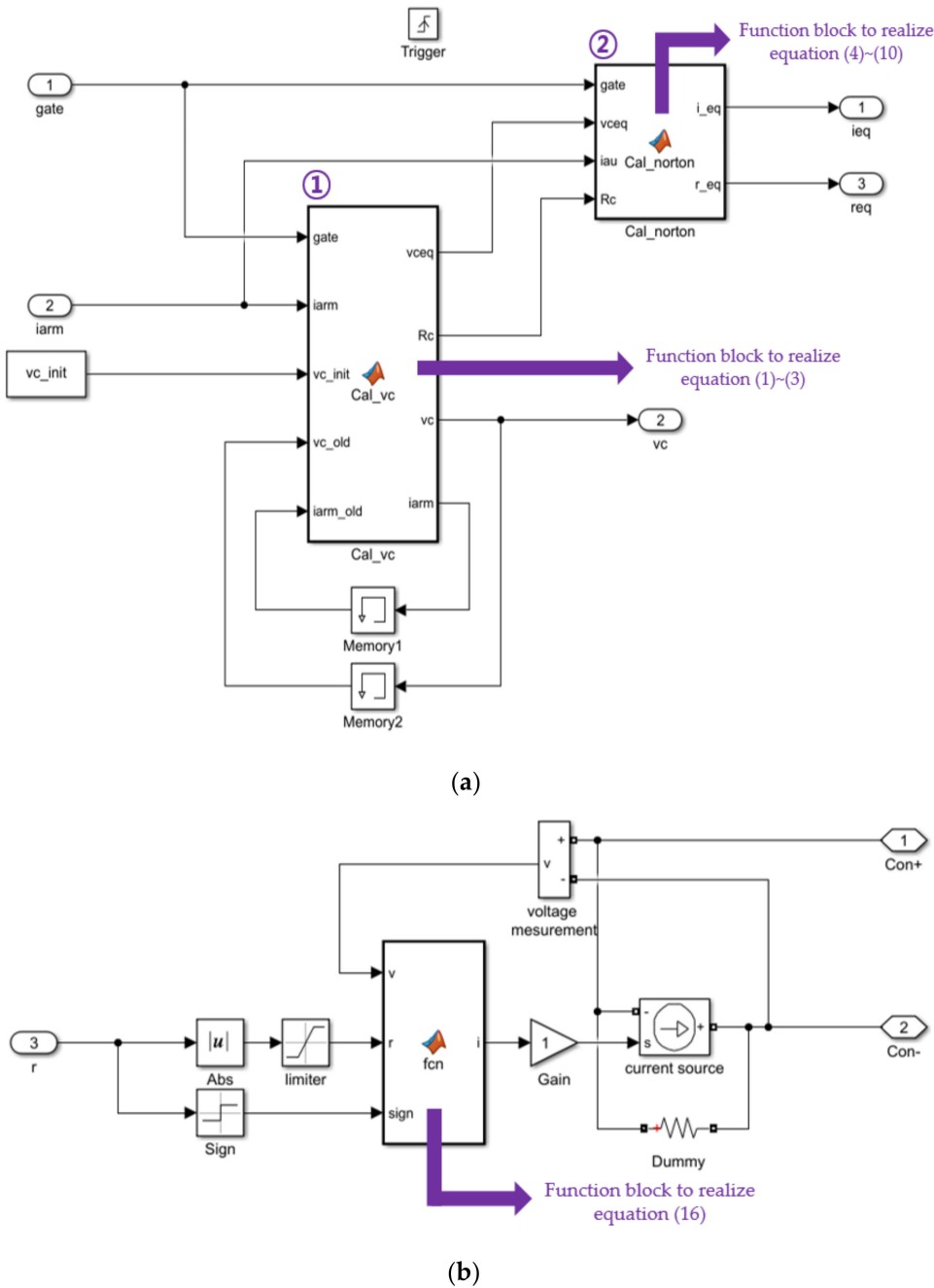

(a)

(b)

**Figure 8.** Implementation block of the subsystems in Figure 7b,c. (**a**) Internal model of the calculating SM valve in Figure 7b; (**b**) Variable resistor implementation block in Figure 7c.

Since the $v_c(t - \Delta T)$ and $i_c(t - \Delta T)$ in Equation (3) require the current result, as well as the calculation result in one cycle ahead, they are made to pass the memory block to give them a delay of one time step, thereby giving the calculation result as an input, which corresponds to the vc_old, iarm_old port in the figure. Through the calculation of Equation (1), the output of Matlab function

block ① has the individual SM capacitor voltages ($v_c$), individual SM output voltages, and the $R_c$ and $v_{cEQ}$ values necessary for Equations (4) and (9) that correspond to the SM valve output voltages that consist of the sum of the foregoing voltages.

For the case of current $i_c(t)$ that flows in the capacitor required in the process of implementing Equations (1) and (3) in Matlab function block ①, as can be seen in Figure 9a, in cases when gate = 1 the top switch $S_1$ of the SM is turned on and the bottom switch $S_2$ is turned off, $i_{arm}(t)$ flows to the capacitor in this state. As can be seen in Figure 9b, in cases when gate = 0 the top switch $S_1$ of the SM is turned off and the bottom switch $S_2$ is turned on, the current in this state becomes 0 as no current flows to the capacitor. Therefore, current $i_c(t)$ can be implemented as the multiplication of the gate value and $i_{arm}(t)$, as expressed by Equations (11) and (12).

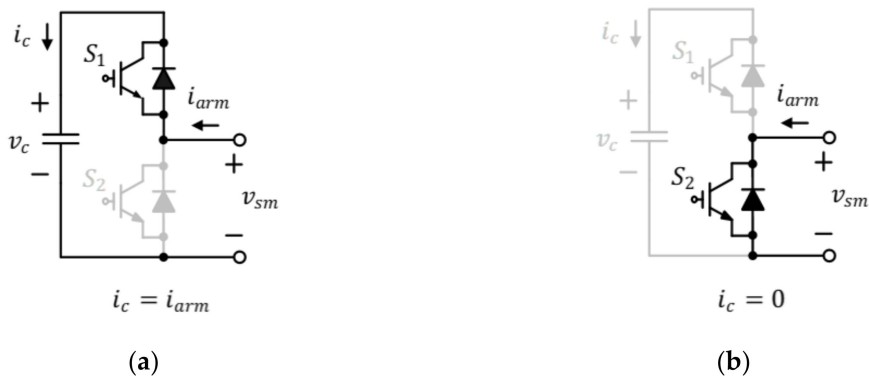

**Figure 9.** Capacitor current flow diagram. (**a**) When the SM is inserted; (**b**) When the SM is bypassed.

$$i_c(t) = gate\ i_{arm}(t) \tag{11}$$

$$i_c(t - \Delta T) = gate\ i_{arm}(t - \Delta T) \tag{12}$$

In function block ①, $v_c(t)$ and $v_{cEQ}(t - \Delta T)$, which correspond to Equations (1) and (3), respectively, are implemented as Equations (13) and (14) using Equations (11) and (12), respectively. Table 1 shows the algorithm of function block ①.

**Table 1.** The process of calculating $R_c, v_c$ and $v_{cEQ}$ by function ① shown in Figure 8a.

| |
|---|
| 1. Get gate, $i_{arm}(t)$, $v_{cEQ}(t - \Delta T)$, and $i_{arm}(t - \Delta T)$ |
| 2. Set N, C and $\Delta T$<br>a) Compute equation (2)<br>b) Compute equation (14)<br>c) Compute equation (13) |
| 3. Output $R_c$, $v_c(t)$, and $v_{cEQ}(t - \Delta T)$ |

$$v_c(t) = R_c\ gate\ i_{arm}(t) + v_{cEQ}(t - \Delta T) \tag{13}$$

$$v_{cEQ}(t - \Delta T) = R_c\ gate\ i_{arm}(t) + v_c(t - \Delta T) \tag{14}$$

Matlab function block ② implements Equation (4) and Equations (7) to (10). The output voltages of individual SMs and the output voltage of the SM valve expressed as the sum of the foregoing output voltages are created. During the implementation of Equation (4), the gate, $v_{cEQ}(t - \Delta T)$, $i_{arm}(t)$, and $R_c$ are received as inputs to implement Equations (5) and (6) and according to gate signals, $R_{on}$

and $R_{off}$ values are calculated using the values of the equivalent resistance $r_1$, $r_2$ of the switches. When the output voltages of the individual SMs have been obtained by Equation (4), the output voltage of the SM valve is obtained by Equation (7) and the $r_{eq}$ and $i_{eq}$ of the SM valve equivalent circuit are obtained as outputs with the calculation processes of Equations (9) and (10). The process of the algorithm is summarized in Table 2.

**Table 2.** Process of calculating $i_{eq}$ and $r_{eq}$ function ② shown in Figure 8a.

| |
|---|
| 1. Get gate, $v_{cEQ}(t - \Delta T)$, $i_{arm}(t)$, $R_c$ |
| 2. Set $R_{on}$ and $R_{off}$<br>*a) if (gate == 1) $r_1 = R_{on}$, $r_1 = R_{off}$*<br>*b) else if (gate == 0) $r_1 = R_{off}$, $r_1 = R_{on}$*<br>*c) Compute equation (5)*<br>*d) Compute equation (6)*<br>*e) Compute equation (4)*<br>*f) Compute equation (7),(8),(9), and (10)* |
| 3. Output $i_{eq}$, $r_{eq}$ |

The resistance $r_{eq}$ of the Norton equivalent circuit is equal to the sum of the resistances ($r_{smEQ}$) of the equivalent circuits of the SMs, as shown in Equations (8) and (9), and $r_{smEQ}$ is composed of the combination of $r_1$ and $r_2$, as shown in Equation (5). In the process of implementation of Table 2, the values of $r_1$ and $r_2$ change according to the gate signals that change in every switching cycle, that is, $r_{eq}$ should be implemented as a variable resistor. It should be mentioned that there is no model of a variable resistor provided by the SIMSCAPE library of the current version of Simulink. Therefore, in this study, a variable resistor was implemented. as shown in Figure 8b. As shown in Equation (15), the variable resistor was implemented with a dependent current source so that the voltages applied to the two ends of the variable resistor can be adjusted according to Ohm's law and the value of the current flowing can be adjusted according to the required resistance value.

$$i = \frac{v}{r} \qquad (15)$$

In the implementation of the variable resistor in this study, the basic idea of Equation (15) was applied with Equation (16).

$$i = \frac{v}{|r|} \times sign(r) \qquad (16)$$

In Figure 8b, the $r$ value of input port 1 is the $r_{eq}$ value obtained at the previous time step. Since the voltage applied to the two ends of the variable resistor should be divided by the value of the variable resistance, as shown in Equation (15), the value of the variable resistance should not be 0. In addition, since a value of 0 means a short circuit, to prevent the value of the variable resistance from becoming 0, an absolute value was taken for $r_{eq}$ by an absolute value block (abs) and the value of 0 was prevented by a limiter block. In this case, the value determined by the limiter block should always be a positive value larger than 0. As can be seen in Equation (5), since the values of $r_1$, $r_2$, *and* $R_c$ are always positive numbers, the value of $r_{smEQ}$ is always a positive value and the value of $r_{eq}$, which is the sum of the foregoing, is also always a positive value. Therefore, the resistance value obtained by the limiter block can be used as it is for the $r$ in Equation (15). However, considering cases where the resistance has a negative value too, so that the value can be used universally, a sign block that can output +1 when the value of $r_{eq}$ is positive and output −1 when the value of $r_{eq}$ is negative was used.

The voltage ($v$) at the two ends of the variable resistor, the resistance value ($|r|$) that passed through the absolute value block and the limiter block, and the value of the sign (*sign*) that passed through the sign block, were made to be received by the Matlab function block as inputs to implement Equation (16) and sent out as a setpoint of the dependent current source. Meanwhile, a dummy resistor with a large resistance value was added in parallel with the current source to prevent simulation errors.

### 3.3. DFIG Wind Power Generation System Model

Figure 10a shows a wind power generation system using DFIG. Figure 10b shows an equivalent model of Figure 10a for HILS used in this study. There are dozens or hundreds of DFIGs in the wind farm, so real-time implementation of this individual generator system using converter and generator models is not possible, since this study aims to use HILS in the development of the MMC's algorithms.

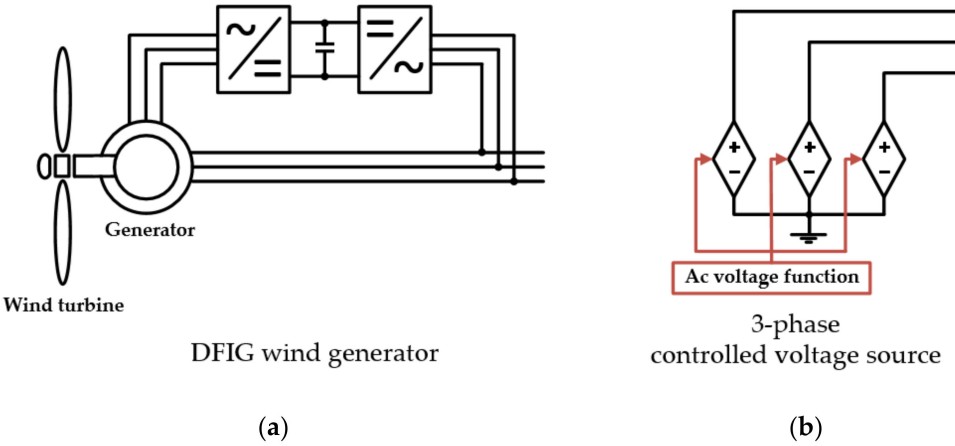

(a)　　　　　　　　　　　　　　　　　　　　　　(b)

**Figure 10.** Doubly fed induction generator (DFIG) wind power generator. (**a**) Configuration of DFIG wind generation system; (**b**) Equivalent model for Figure 10a.

Wind power generation units are developed as simplified models that can generate changes in active and reactive power depending on the wind speed for use in the HILS application. To simulate the operability of wind power generation and also perform the operation of the maximum power point tracking (MPPT) according to wind speed, it is modeled using a three-phase dependent voltage source that can vary the phase angle and magnitude of the output voltage. When interpreting the sending and the receiving ends, which become the targets of interest to control the active power, which is the amount of wind power generated, and the reactive power, they can be expressed as a four-terminal circuit with the voltage ($\widetilde{E}_S$) and current ($\widetilde{I}_S$) of the sending end and the voltage ($\widetilde{E}_R$) and current ($\widetilde{I}_R$) of the receiving end, as shown in Figure 11. When expressed as a series impedance circuit, they can be expressed as a matrix where the voltage and current of the sending end and those of the receiving end are expressed between the sending end and receiving end as Equation (17), where X is the impedance value including the transmission line and the transformer.

$$\begin{bmatrix} \widetilde{E}_S \\ \widetilde{I}_S \end{bmatrix} = \begin{bmatrix} 1 & X \\ 0 & 1 \end{bmatrix} \begin{bmatrix} \widetilde{E}_R \\ \widetilde{I}_R \end{bmatrix} \tag{17}$$

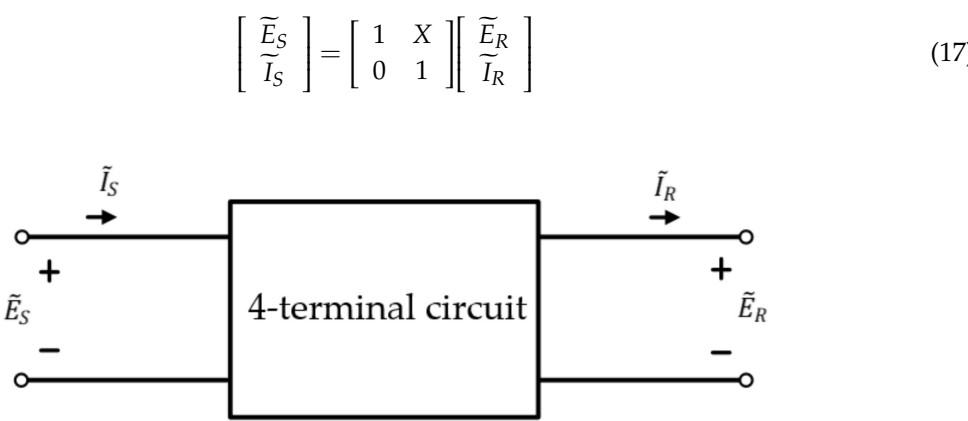

**Figure 11.** Four-terminal circuit to show the relation of power transmission.

The current of the sending end as shown by Equation (18) can be obtained from Equation (17) and the complex power at the sending end can be obtained using the current of the sending end,

as shown by Equation (19), where $\delta$ is the phase difference between the sending end voltage and the receiving end voltage. With the foregoing, the active power and reactive power at the sending end can be obtained as Equations (20) and (21), respectively.

$$\widetilde{I}_S = \widetilde{I}_R = \frac{\widetilde{E}_S - \widetilde{E}_R}{jX} \tag{18}$$

$$S_S = P_S + Q_S = \widetilde{E}_S \widetilde{I}_S^* = E_s \left( \frac{\widetilde{E}_S - \widetilde{E}_R}{jX} \right)^* \tag{19}$$

$$P_s = \frac{E_S E_R}{X} \sin \delta \tag{20}$$

$$Q_s = \frac{E_S{}^2 - E_S E_R \cos \delta}{X} \tag{21}$$

When the foregoing is applied to the modeling process, the sending end corresponds to a wind farm and the receiving end corresponds to the alternating current side of the WSM (Wind power generator Side MMC). $E_r$ is WSM output voltage, so it is a known value that knows the phase and its magnitude. Therefore, by implementing the magnitude and phase of $E_s$ through the dependent voltage source, the amount of active power and reactive power of Equations (20) and (21) can be adjusted. The amount of active power generated is determined from the MPPT curve corresponding to the wind speed. This amount was determined through the table of wind speed versus active power, and the reactive power of the generator was calculated to be a power factor of 0.9.

## 4. HILS Implementation

Figure 12 shows an overall schematic diagram of the HILS system applied to the experiment. The HILS is a combination of a real-time OS and hardware. The real-time OS in this study is OP4510 from Realtime Wave company. The Simulink model of the MMC implemented via DEM and the wind power generation block modeled using dependent current are compiled using the RT-LAB tool and performed at OP4510. Data exchange between the PC and the real-time OS takes place by Ethernet communication. In addition, the hardware corresponding to the HILS is a controller composed of FPGAs. The interface between OP4510 and the FPGA boards is achieved with the analog I/O and digital I/O of OP4510. Input/output allocation is done at RT-Lab tool compilation. Through the analog output (AO) of OP4510, the voltage value of the MMC is transferred to the external controller, and digital input (DI) is used to receive the command from the external controller, an FPGA board.

The experimental setup is shown in Figure 13. The MMC and wind power generation model proposed in this study are executed in ① OP4510. The FPGA controller is ⑤. Since the input/output voltage level of the FPGA's GPIO is 3.3 V, and the digital input/output port of OP4510 requires a voltage of at least 5 V, the voltage level change board ② works for that. Circuit ③ generates 3.3 V and ④ is a 5 V DC-DC converter. Table 3 summarizes the parameters of the MMC system used for both PC-based simulation and real-time simulation. The amounts of wind power generated were set so that 500 W would be delivered when wind speeds are 7~15 m/s and 700 W would be delivered when wind speeds are 15~20 m/s, and the power factor of DFIG was set to 0.9. The amounts of reactive power and active power to that end are obtained by adjusting the magnitude and phase angle of the voltage of the dependent current source followed by Equations (20) and (21).

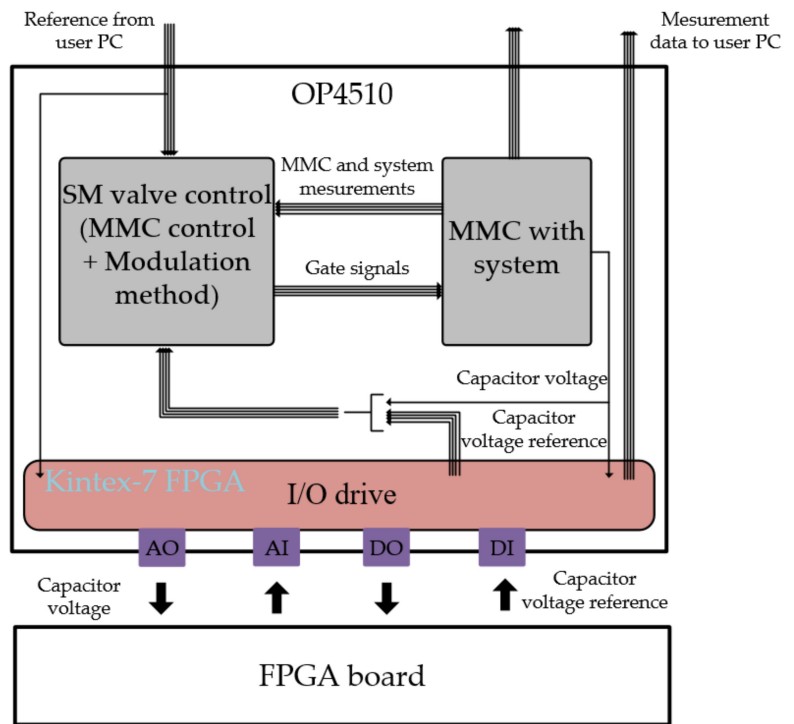

**Figure 12.** Overall configuration of the HILS.

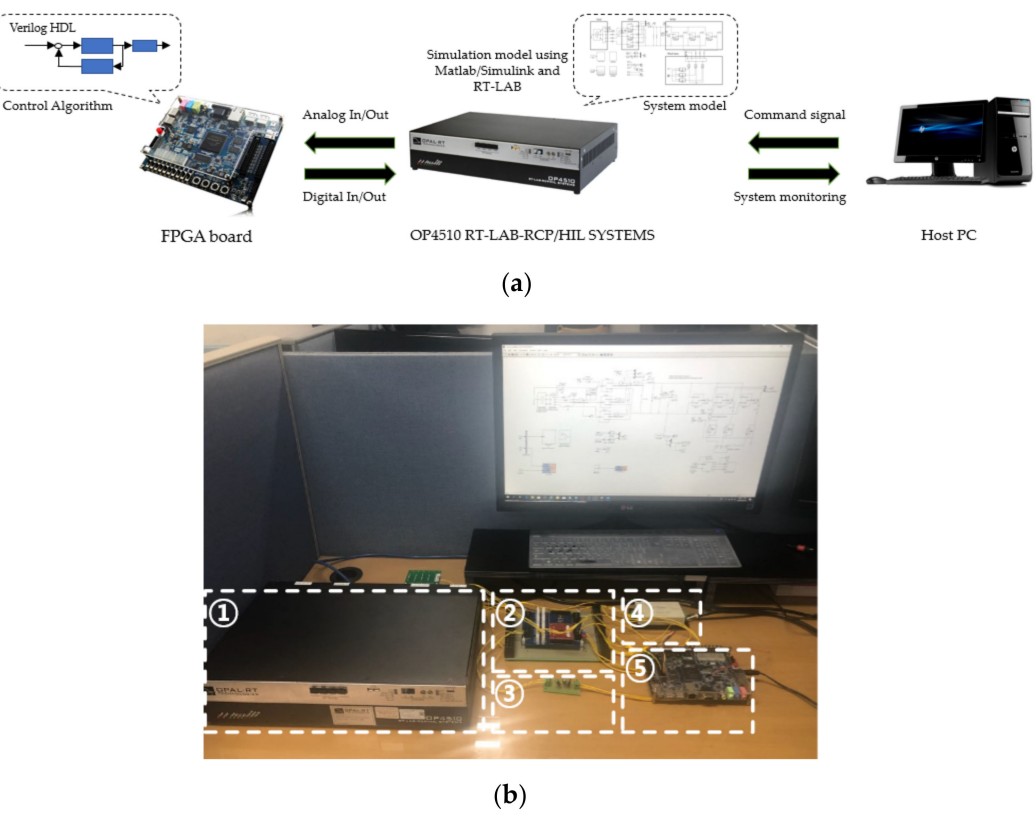

**Figure 13.** Experimental setup. (**a**) Conceptual diagram; (**b**) Picture.

**Table 3.** Parameters of the MMC system.

| Quantity | Value | Comment |
|:---:|:---:|:---:|
| PF | 1 | Power factor at grid side |
| $L_{arm}$ | 3 mH | Inductance of arm inductor |
| $R_{arm}$ | 0.5 Ω | Resistance of arm inductor |
| $C_{sm}$ | 3 mF | SM capacitance |
| $L_a$ | 1 mH | Grid side inductance |
| $v_{a\phi}$ | 220 V$_{rms}$ | Grid side phase voltage |
| $V_{dc}$ | 700 V | DC side rated voltage |
| $C_{dc}$ | 7.5 mF | DC side capacitance |
| $f_s$ | 60 Hz | Electric frequency |
| $N$ | 30 | Number of SM per arm |

*Voltage Controller of the MMC by the FPGA*

In this section, the FPGA, which is an external controller, is explained. The DE1-SoC board from the company Terasic was used in this study, which employs Altera Cyclone®5CSEMA5F31C6N. The controller's role is voltage control, and the corresponding area of the designed controller is shown in Figure 14.

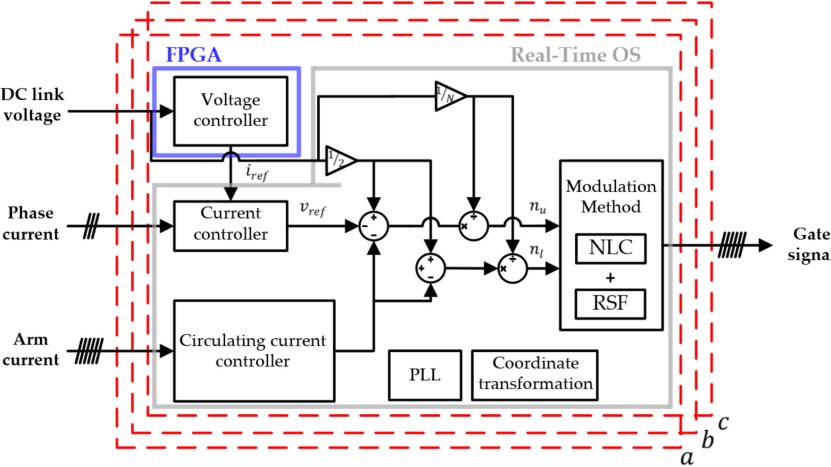

**Figure 14.** Whole control configuration for the MMC including the FPGA controller.

To use ADCs (Analog to Digital Converters) that are in charge of data input/output as with voltage controllers and peripheral devices such as GPIO (General Purpose Input Output), the implementation of an FSM (Finite State Machine), which is widely used in the configuration of control circuits of digital systems, is essential. FSM is an abbreviation for finite state machine, which means a machine with limited states. An FSM is used when designing controllers that play the role of the brain of the hardware to determine how the hardware should operate based on changes in limited states. The basic structure of a FSM can be implemented with three hardware blocks, as depicted in Figure 15. These blocks include a block (state logic) that determines the next state according to the input and current state. The next state output by this block is input into the state register, and when the clock signals coming into the state register become a positive edge, the next state is output as the current state. The output logic receives the current states and input signals as inputs to determine outputs.

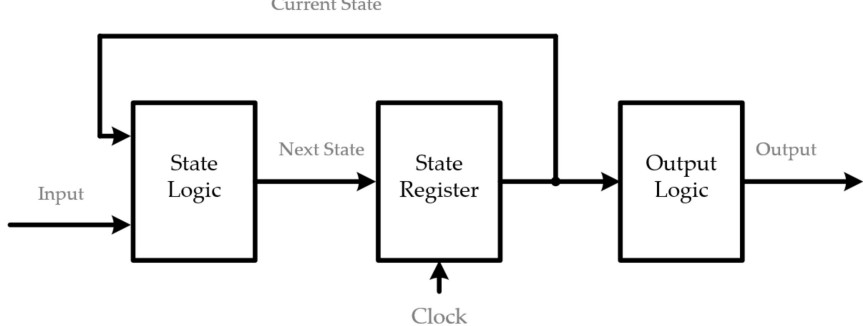

**Figure 15.** Basic block diagram of an FSM.

A state machine diagram of the controller designed with the basic structure of the FSM is shown in Figure 16. The state machine has a total of four states, i.e., stop state, memory state, calculation state, and output state. The stop state is whether the controller waits for the operation, and the state is changed to the memory state when the operation starts with the external switch. In the memory state, the DC terminal voltage is transformed from analog into a 12-bit digital form through the ADC and the transformed data are stored. When storage is completed, the state is changed into the next state, i.e., the calculation state. In the calculation state, the data stored in the memory state are received as inputs to operate the PI controller. The output of the PI controller is as shown by Equation (22), and to implement it as embedded codes, Equation (24), which corresponds to the backward transform for digital expression, can be applied to Equation (23) to write it as shown by Equation (25), which is implemented in the form of Equation (26). Where, $x(s)$ and $x^*(s)$ correspond to the DC stage voltage and its command value, respectively. In this case, $y(s)$ is the current command for DC side voltage control.

$$y(s) = \left(K_p + \frac{K_i}{s}\right)(x^*(s) - x(s)) \tag{22}$$

$$= \left(K_p + \frac{K_i}{s}\right)e(s) \tag{23}$$

$$s \rightarrow \frac{1 - z^{-1}}{T} \tag{24}$$

$$\left(\frac{1 - z^{-1}}{T}\right)y(z) = K_p\left(\frac{1 - z^{-1}}{T}\right)e(z) + K_i e(z) \tag{25}$$

$$y(n) = y(n-1) + K_p(e(n) - e(n-1)) + K_i Te(n) \tag{26}$$

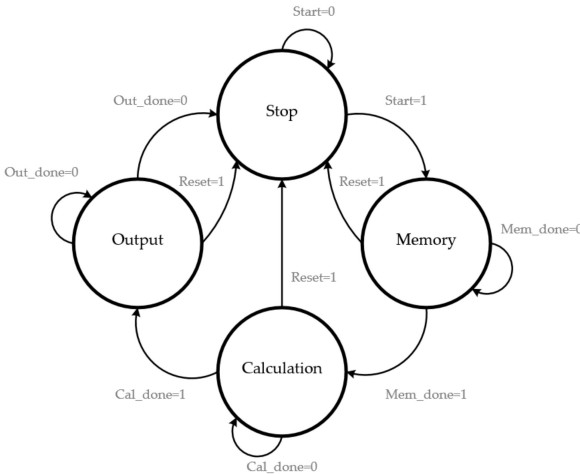

**Figure 16.** State machine diagram of the voltage controller in the FPGA.

When the calculation is completed, the state is changed into the output state. In the output state, the output of the calculation state is received as inputs to update the data value transmitted to the real-time OS through the 2 x 20 pin GPIO.

The resources used in the controller design are shown in Table 4. The main CLOCK frequency of the FPGA is 50 MHz, and the time to read and store ADC values is taken by LTC2308 for a total of 2 μs, and it takes 200 ns with 10 clocks to update the computation and output data. It is equivalent to 2.2% when the voltage controller cycle is assumed to be 100 μs and has a higher bandwidth than a typical DSP-based controller.

**Table 4.** Resources used by the FPGA (CycloneV, 5CSEMA5F31C6).

| Resource | Use/Total |
| :---: | :---: |
| Logic utilization | $1,836/32,070$ |
| Total registers | 408 |
| Total pins | 212/457 |
| Total DSP blocks | 6/87 |
| Total PLLs | 1/6 |

## 5. Experimental Results of the PC-Based Simulation and from the HILS

This section shows simulations performed on PCs and real-time experiments of HILS systems configured with the proposed model-applied real-time OS and FPGA controllers. Figures 17 and 18 illustrate waveforms from a Simulink simulation, and the real-time HILS system, respectively. In other words, Figures 17 and 18 show the waveforms in the grid-side MMC performed on the PC and HILS, respectively. If mentioned in advance, Figure 18, the experimental results, and Figure 17, the simulation results operated on the PC, are in agreement.

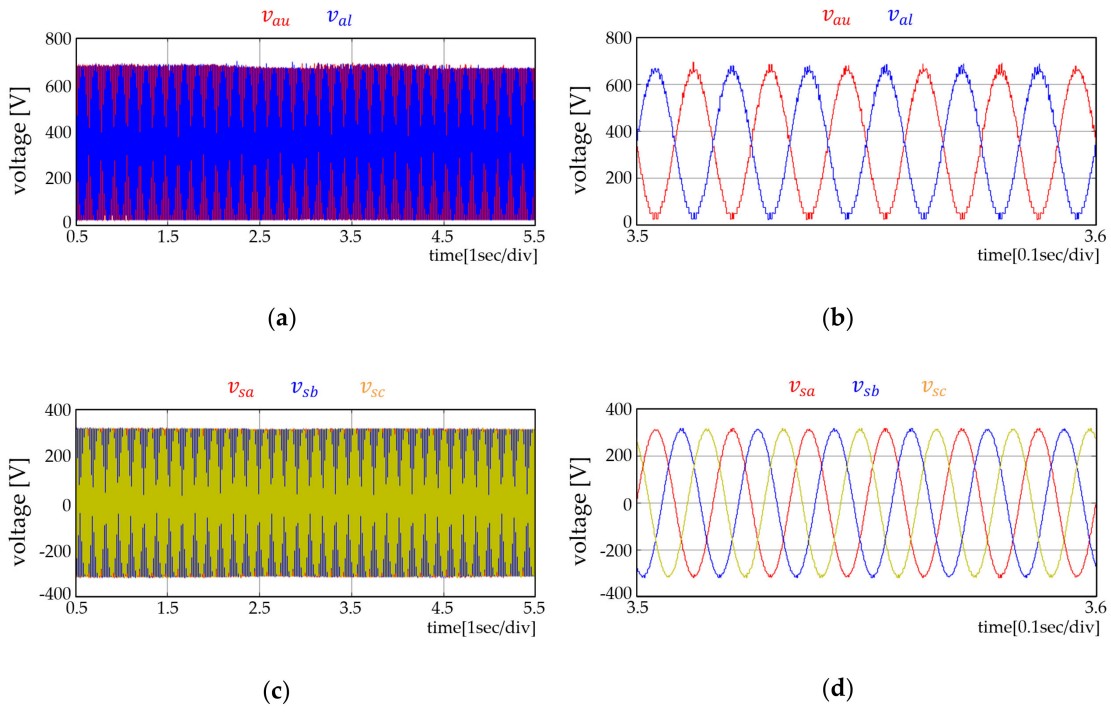

**Figure 17.** *Cont.*

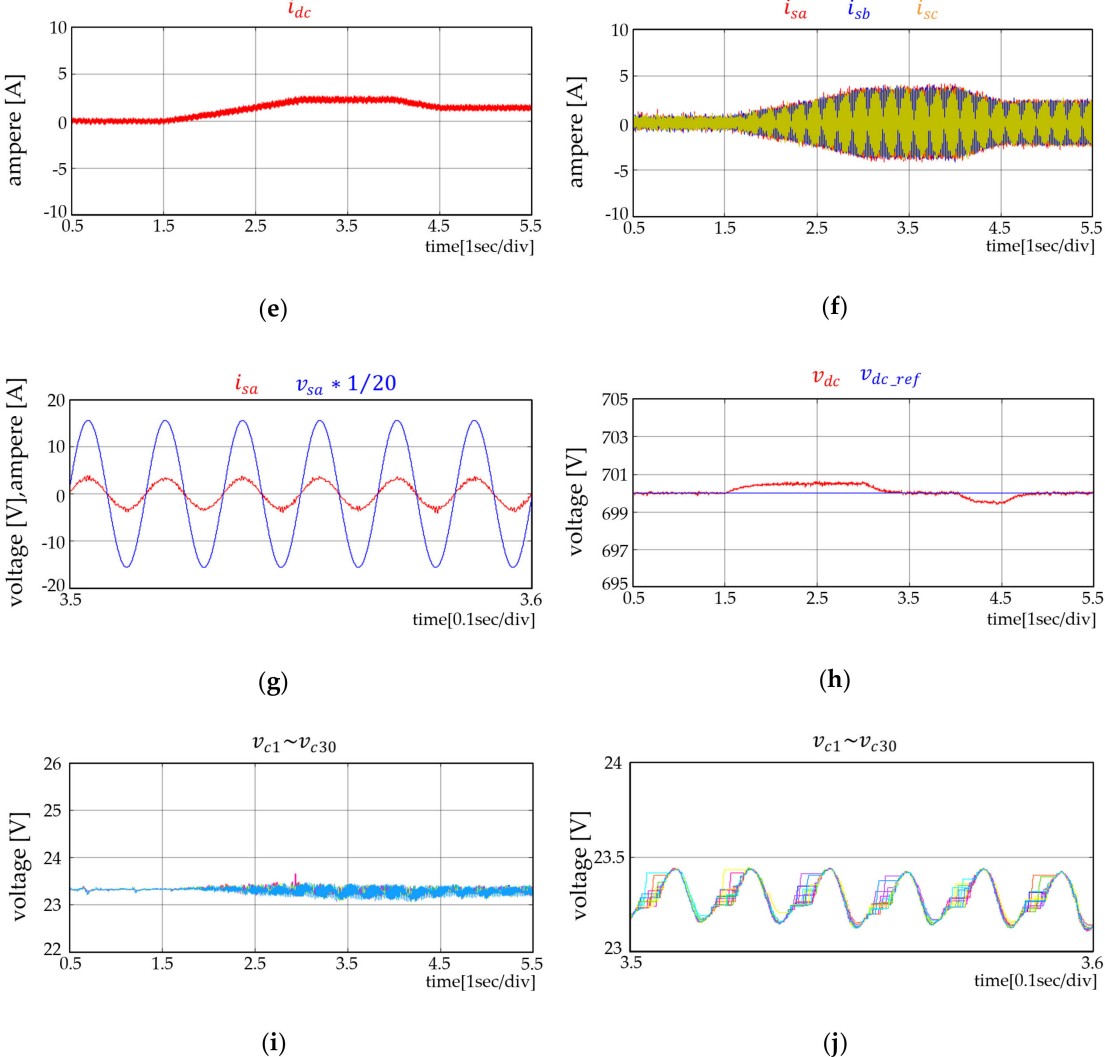

**Figure 17.** Simulation results performed on a PC. (**a**) Arm voltage of phase A; (**b**) Enlarged arm voltage of phase A; (**c**) Pole voltage; (**d**) Enlarged pole voltage; (**e**) DC side inflow current; (**f**) Phase current; (**g**) Phase current and pole voltage of phase A; (**h**) DC side voltage and its reference; (**i**) Capacitor voltages of upper arm SM on phase A; and (**j**) Enlarged capacitor voltages of upper arm SM on phase A.

Whereas the implementation of real-time simulations in the HILS experiments is different from the Simulink simulations carried out only on a PC, to implement the HILS, the configuration of simulations should be divided into two areas, that is, a master subsystem in charge of the implementation of simulations and operations such as data storage, and a console subsystem that observes data and delivers setpoints. The master subsystem is implemented in the real-time OS for real-time simulations and the console subsystem observes data in real time and revises setpoints in the user's PC. The data on the simulations carried out with Simulink are displayed using the scope in the Simulink. The HILS experiments are carried out in the real-time OS and the stored data are displayed using the Scopeview of the RT-LAB tool.

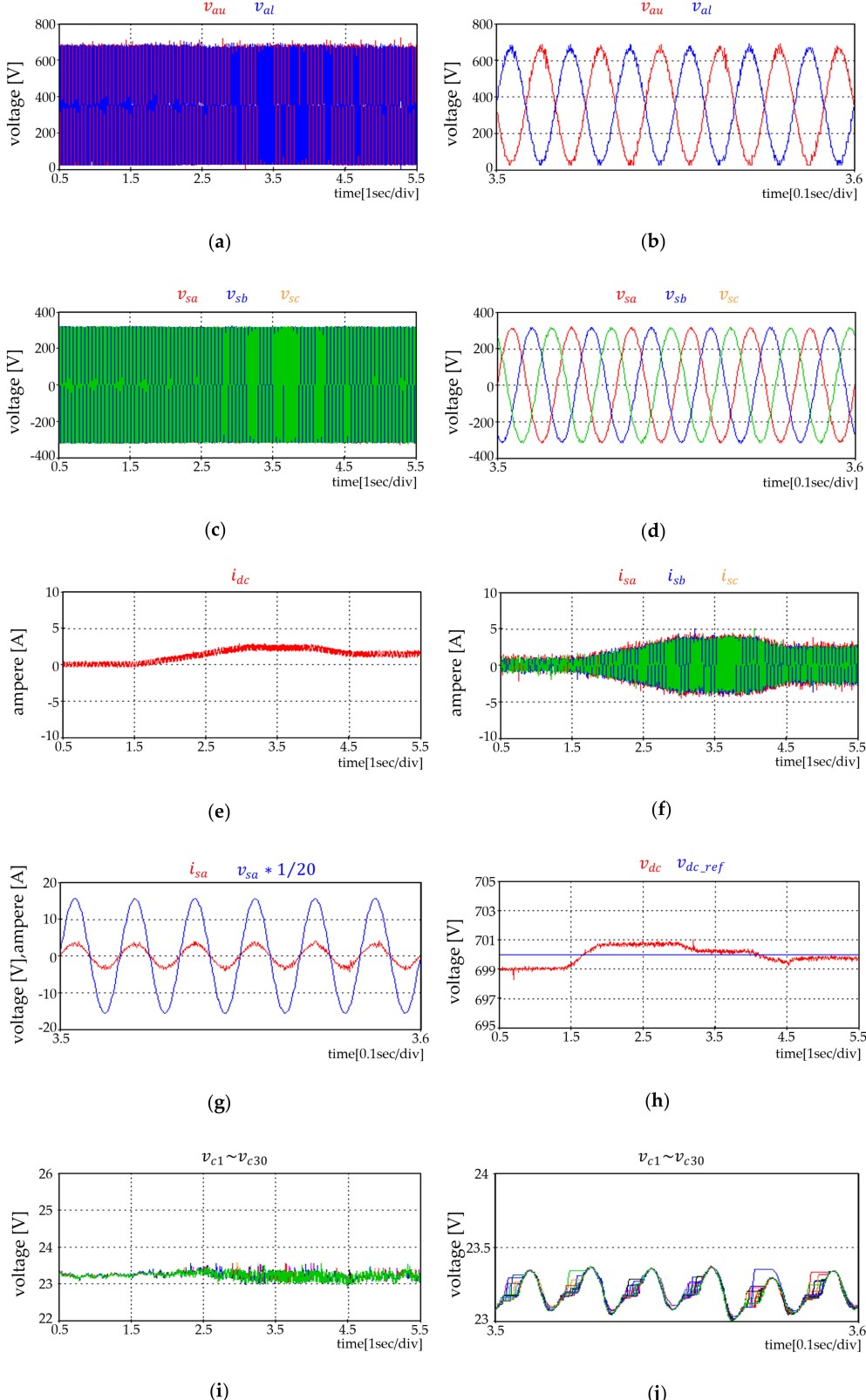

**Figure 18.** Experimental results using the proposed HILS system. (**a**) Arm voltage of phase A; (**b**) Enlarged Arm voltage of phase A; (**c**) Pole voltage; (**d**) Enlarged pole voltage; (**e**) DC side inflow current; (**f**) Phase current; (**g**) Phase current and pole voltage of phase A; (**h**) DC side voltage and reference; (**i**) Capacitor voltages of upper arm SM on phase A; and (**j**) Enlarged capacitor voltages of upper arm SM on phase A.

It is intended to show the transfer ability of generated power to grid and control capability of the DC-link voltage with regard to wind speed change. The change of wind speed is as follows: Starting from 0.5 seconds after DC stage voltage stabilizes, the wind speed reaches zero to cut-in speed at 1.5 seconds and the power generation stays at zero watts; then, from 1.5 seconds to 2.5 seconds, the wind speed reaches the rated wind speed, increases the power generation until t = 3.5 seconds, and the wind speed drops to the reduced power generation until t = 4 seconds; then, maintaining a constant wind speed after t = 4 seconds. The arm voltages, as shown in Figure 17a,b, are balanced with 180 degrees inversion each other, and the pole voltages, shown in Figure 17c,d, are also three-phase balanced. Figure 17e illustrates the DC stage current flowing from WSM through the DC stage to GSM (Grid Side MMC), and by compared with the grid-side current, shown in Figure 17f, the change in power generation can be confirmed.

Figure 17g depicts the pole voltage and phase current, and it can be seen that the power factor at the grid side is controlled to 1 by confirming that the voltage and current waveforms are in phase. Figure 17h shows the DC stage voltage and the setpoint, and the DC stage voltage has a ripple of 0.5 V, which corresponds to about 0.07% of the setpoint, so it can be seen that the voltage control is well done. Figure 17i,j shows the SM capacitor voltage of the phase A upper arm, balanced at 23.33 V, which is the value of the DC stage value divided by the number of SMs per arm.

Figure 18 depicts the results from the HILS system with the proposed model and the control board. The experimental results are displayed using Scopeview. The experimental results shown in Figure 18 agree with those shown in Figure 17, which are the simulation results. Since these are simulations of the same conditions as Figure 17, the descriptions of individual waveforms are the same as those of Figure 17. The feasibility of the proposed system model can be said to be proven, as shown in Figure 18, where, as compared with the case of Figure 17, the slight differences in the DC stage voltage and the SM capacitor voltage are observed. The ripple of the DC stage voltage, shown in Figure 18h, is 1.6 V, which is about 0.23% deviation from the setpoint. In comparison, the error in Figure 17h, which is from the PC-performed model, is 0.07%. The reason is thought to be due to the precision of numbers because the voltage control, shown in Figure 18, is performed on the FPGA, the external controller. Meanwhile, the SM capacitor voltage depicted in Figure 18j also has a maximum difference of 0.16 V as compared with that depicted in Figure 17j, which is a very small value with an error of 0.7% for the 23.33 V SM voltage.

Real-time implementation can be verified by the number of overruns. In this experiment, the number of overruns showing the real-time performance was zero, so running in real time was confirmed. The proposed HILS model spent 1 second in 1 second real-time simulation, whereas the simulation on the PC using the Simulink model took 118 seconds.

## 6. Conclusions

Since sizeable costs and time are invested when test environments are actually made to configure MMC and AC systems when a controller for MMCs is developed, it is evident that HILS systems should be applied. To this end, test environments based on accurate modeling are necessary. In this study, a real-time operation model of MMCs applied with DEM using Simulink was proposed. In addition, we introduced a method of equivalent modeling using a wind power generation system as a dependent current source, showed the process of controller design using FPGA, and implemented a HILS system that combined these two systems. To implement the equivalent circuit of the MMC module, the equivalent resistors according to switching states were calculated and a method to implement variable resistance for the creation of the output voltage of the MMC arm was explained. The results of simulations implemented in the PC, which took 118 seconds for the implementation of 1 second, and the results of the HILS experiments were compared to show that the results were identical. As a result of the foregoing, the validity of the proposed model was demonstrated. In addition, during the HILS experiment, a controller using FPGA was used to derive the results. Since the proposed MMC is based on equivalent circuits, the increase and decrease in the level of the output voltage has some

degree of freedom. The MMC proposed in this study can easily be expanded to 31 levels of MMCs connected back-to-back or several hundred levels of MMCs for connections among grids. In addition, since this model was developed using Simulink, if the various libraries provided by Simulink are grafted, its usefulness can become even greater.

**Author Contributions:** Conceptualization, D.-M.L. and D.-C.S.; Funding acquisition, D.-M.L.; Methodology, D.-M.L.; Software, D.-C.S. and D.-M.L.; Validation, D.-C.S. and D.-M.L.; Writing—Original draft preparation, D.-C.S. and D.-M.L.; Writing—Review and editing, D.-M.L. and D.-C.S.; Visualization, D.-C.S.; Supervision, D.-M.L. All authors have read and agreed to the published version of manuscript.

**Funding:** This research was supported by the Korea Electric Power Corporation (grant number: R17XA05-18).

**Conflicts of Interest:** The authors declare no conflict of interest.

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
