# Peer review of "Development of Real-Time Implementation of a Wind Power Generation System with Modular Multilevel Converters for Hardware in the Loop Simulation Using MATLAB/Simulink"

_electronics, doi:10.3390/electronics9040606_

Round 1

Reviewer 1 Report

Below suggestions for improvement:

  1. Title: avoid using acronyms in the title. Not all readership is aware of the meaning of such acronyms. Consider an alternative (longer) title instead.
  2. Define acronyms before use them (e.g., PSCAD/EMTDC). Make edits throughout the manuscript. Do not define acronyms in the abstract, instead, write the full meaning. Defining acronyms in the introduction section is more appropriate.
  3. The Authors need to seek assistance from a native English speaker, as the article is difficult to read. Connecting ideas are not clearly presented or the arguments are too vague. Also, it is recommended that phrases such "as mentioned earlier" be removed throughout the manuscript.
  4. Introduction section: The phase: "In particular, wind power generation is attracting attention because it is less burdensome to maintain and provides clean energy" is completely inaccurate. The real reason behind wind energy expansion in recent years is due to its significant lower levelized cost of energy. Please make appropriate citations the global wind energy council (GWEC) or the U.S. American Wind Energy Association (AWEA) to back up your statements and provide reasonable justification to your work.
  5. Introduction section: nowadays, most wind farms operate with a HVAC topology because the grid-wind farm interconnection point is within 100 Km, and only a handful of offshore wind farms have implemented a HVDC topology (see "A Review of Methodological Approaches for the Design and Optimization of Wind Farms" published in MDPI Energies for a brief description on this matter). Thus, Authors need to elaborate much more on why this study is expected to provide a significant impact on the power-electronics and control of wind farms. Moreover, it is important to note that Joule-based losses are usually more important than losses associated to reactive power generation/consumption, which are typically handled with capacitor banks. Hence, Authors need to formulate convincing statements that  properly justify the work.
  6. Section 3: Rearrange equations 7 and 10. Equations seem to be disorganized in this draft.
  7. Figure 6: it is impossible to read the labels and follow the discussion presented by the Authors. Consider using a complete (rotated) page to show this figure, as it is critical for the work.
  8. Increase the font size of Figure 7a.
  9. Experiments: some assumptions for the experimental component are not reasonable enough. For example; the amount of power generated by the "wind farm" as a function of wind speed was discretized in 3 levels and a constant power factor of 0.9 was assumed. In reality, the power conversion of a single wind turbine depends on the cube of wind speed: P = 0.5*rho*A*v^3*cp*ce, where rho is the air density, A is the swept area of the rotor (pi*R^2, where R is the rotor radius), v is the horizontal component of the wind velocity vector, cp is the power coefficient (it represents the aero-electromechanical properties of the rotor and has a theoretical maximum of 16/27 given by the Betz limit. Typical values range between 0.3 and 0.5 for large-scale wind turbines), and ce is the mechanical-to-electrical power conversion factor, which typically ranges from 0.8 to 0.99. Authors are encouraged to conduct experiments considering more representative physics of wind power conversion.
  10. Rearrange equations 22 and 23, it is not clear what terms belong to which equations.
  11. Figure 18, it is difficult to see differences in the voltages of figure 18a and 18b, consider using an insert such as in figure 18g.
  12. Section 5: The comparison of the HILS system and the proposed model is qualitative only. To better understand the benefits of the proposed model, an in-depth "quantitative" comparison is needed. This involves conducting an error analysis and showing points where the proposed model disagrees with the HILS system. As of now, the qualitative discussion of the agreement/disagreement is very vague. There is no need to create a "table 5" with comparisons of computing times, the description provided at the end of section 5 is sufficient.

Author Response

The authors sincerely appreciate all your valuable comments and suggestions.

Reviewer 2 Report

The presentation and discussion of the simulation results is not satisfactory. In the opinion of this reviewer, the results must be presented in conjunction with the performance of the controllers. The examples used for demonstration, such as wind speed changing from 0 to 7 m/s in 0.2 s, from7 to 15 m/sec in 0.3 s and finally from 15 to 20 m/sec in 0.3325 s do not tally with the reality and have no practical relevance. The instantaneous values of the variables presented in Fig. 18 and Fig. 19 are in and of themselves not that informative. For one, the results obtained using standard SIMULINK model and the HILS system are drawn separately. As a result only the shape of the curves can be compared to one another (and not the results themselves). Secondly, it is difficult to assess the performance of the controllers as no controller set-point changes are included.

I recommend that the paper be improved with emphasis on the presentation of the results by including realistic test-cases, so that a meaningful assessment of the capability of the model can be evaluated.

Author Response

(The authors gave the same response as above.)

Reviewer 3 Report

The paper interesting and timely regarding the topic. However, it is rather hard to follow, starting by the title that has acronyms, which should be avoided. Also, the scientific contribution of the paper is not clear and well justified in comparison with previous literature references.

In the results, HIL concept is used, which is a major plus. However, it is not highlighted in the abstract for example, which makes the paper inconsistent.

The information in Fig. 17 is not needed.

The comparing of results with other approaches is deserved. Authors compare HIL with simple windows implementation but it is a comparison of the model with itself running in a different machine, lets say.

Author Response

(The authors gave the same response as above.)

Reviewer 4 Report

Most papers discuss either aerodynamics/structures aspect of wind energy or grid etc. However, in this paper the authors have touched upon the link between the grid and the turbine. This is very unique to consider the application of the MMC  that connects wind power to a grid through high-voltage, direct current (HVDC). 

Author Response

<Comment>

Most papers discuss either aerodynamics/structures aspect of wind energy or grid etc. However, in this paper the authors have touched upon the link between the grid and the turbine. This is very unique to consider the application of the MMC  that connects wind power to a grid through high-voltage, direct current (HVDC). 

<Answer>

The authors really appreciate your comment and your time for reviewing this paper.

Round 2

Reviewer 1 Report

Upon reviewing the rebuttal letter provided by the Authors and analyzing their new draft, I noticed that 4 of 12 of my previous recommendations were unsatisfactorily addressed. Below a summary:

  1. In my previous review, I highlighted that wind power development is driven by the tecno-economic benefits that it provides, rather than “because is less burdensome to maintain and provides clean energy”. It is true that wind energy conversion produces no emissions (during operation), but this benefit is not the main driver. Nowadays, the cost of wind energy is about the same as the cost of the energy generated with state-of-the-art combined cycles or gas power plants. Authors were encouraged to cite reports of the Global Wind Energy Council (GWEC) to direct the readership to detailed discussions on the levelized cost of energy. The Authors expressed in their rebuttal letter that they included such references in their revised manuscript, but I cannot see these. Moreover, in their rephased sentence “Wind power generation has the advantage of using vast resources of wind power, and attracts attention as clean energy that can minimize the emission of harmful substances [1-4]”, references 1 to 4 do not deal with “the advantage of using vast resources of wind power” or “how clean energy minimize the emission of harmful substances”.
  2. In my previous review, I highlighted that HVDC topologies for wind farms are very abnormal, and only used in few distant offshore wind farms. I requested the Authors to elaborate on why they think their work (based on HVDC topologies) would produce an impact in wind farms. The Authors provided an argument that does not deal with the aforementioned point at all. In a nutshell, the Authors responded that the significance of the paper is based on the observation that no one has done what they proposed.
  3. In my previous review, I highlighted that the assumptions of the experimental component are not reasonable enough, and the Authors acknowledged this observation in their rebuttal letter. Considering changing wind speeds in discretized steps and modeling wind power generation using a voltage source is still unreasonable and does not reflect the reality in which wind farms operate. Please consider that electrical power depends on the cube of wind speed, thus, for doubling wind speed, electrical power is increased 8 times. If Authors are just testing their model (as indicated by them: “this is a data tor order to show the validity of the proposed model”), then, they should present the experiments as “a generic example” that is not associated to the real context of wind power conversion in wind farms, as it is misleading and can generate incorrect interpretations by the readership. Moreover, the explanations provided in the rebuttal letter about the power curve and the power conversion process of the wind turbine are not included in the revised manuscript.
  4. I still don’t understand equation 23, there are two equations overlapped in the same line.

Finally, the paragraph defined on lines 405 to 417 should be rephrased, Authors indicate that the feasibility of the proposed model is proven by just observing figures 18 and 19, and then they present a simple error analysis. In general, the error analysis determines whether the proposed model is feasible or not. Thus, the order of ideas should be reconsidered.

Reviewer 2 Report

The paper can be published in the present form

Reviewer 3 Report

My opinion is that the revised version is at the same level of the initial one.

going back to previous comments:

  • Also, the scientific contribution of the paper is not clear and well justified in comparison with previous literature references.

-> the authors say now that "it is difficult to find a literature". This is not the acceptable way to state the contribution. and it reflects that in the rest of the paper, the scientific contribution is not clear or at this journal level.

  • The information in Fig. 17 is not needed.

-> this reviewer insist that this figure has no value. it is a well know and mandatory aspects of OPAL target. The authors dont agree.

  • The comparing of results with other approaches is deserved. Authors compare HIL with simple windows implementation but it is a comparison of the model with itself running in a different machine, lets say.

-> more simulations were added but no comparison with other works in the literature